# Molecular and Metabolic Insights into Anthocyanin Biosynthesis for Spot Formation on *Lilium leichtlinii* var. *maximowiczii* Flower Petals

**DOI:** 10.3390/ijms24031844

**Published:** 2023-01-17

**Authors:** Zhen Wang, Xin Li, Minmin Chen, Liuyan Yang, Yongchun Zhang

**Affiliations:** Forest & Fruit Tree Institute, Shanghai Academy of Agricultural Sciences, Shanghai 201499, China

**Keywords:** *Lilium leichtlinii* var. *maximowiczii*, petal spots, anthocyanin biosynthesis, transcriptomics, metabolomics

## Abstract

Plants exhibit remarkable diversity in their petal colors through biosynthesis and the accumulation of various pigments. *Lilium*, an important cut and potted flower, has many coloring pattern variations, including bicolors and spots. To elucidate the mechanisms regulating spot formation in *Lilium leichtlinii* var. *maximowiczii* petals, we used multiple approaches to investigate the changes in petal carotenoids, spot anthocyanins, and gene expression dynamics. This included green petals without spots (D1-Pe and D1-Sp), yellow–green petals with purple spots (D2-Pe and D2-Sp), light-orange petals with dark-purple spots (D3-Pe and D3-Sp), and orange petals with dark-purple spots (D4-Pe and D4-Sp). D3-Pe and D4-Pe contained large amounts of capsanthin and capsorubin and small amounts of zeaxanthin and violaxanthin, which contributed to the orange color. In addition to cyanidin-3-O-glucoside, pelargonidin-3-O-rutinoside, cyanidin-3-O-rutinoside, and peonidin-3-O-rutinoside may also contribute to *L. leichtlinii* var. *maximowiczii*‘s petal spot colors. KEGs involved in flavonoid biosyntheses, such as *CHS*, *DFR*, and *MYB12*, were significantly upregulated in D2-Sp and D3-Sp, compared with D1-Sp, as well as in spots, compared with petals. Upregulated anthocyanin concentrations and biosynthesis-related genes promoted spot formation and color transition. Our results provide global insight into pigment accumulation and the regulatory mechanisms underlying spot formation during flower development in *L. leichtlinii* var. *maximowiczii*.

## 1. Introduction

Color is a visible trait of plants and is greatly significant to plant growth and development. Flower color formation relies on the type and content of pigment. In most plants, flower colors are mainly determined by flavonoids, especially anthocyanins, carotenoids, and betalains [1]. Flavones, flavanones, chalcones, and other compounds in flavonoids form yellow colors. Anthocyanins form red, purple, and blue colors. Carotenoids are yellow, orange, and red. Betalains form yellow, orange, and red colors, but they only exist in some Caryophyllaceae plants and cannot coexist with anthocyanins [2].

Anthocyanin biosynthesis is a branch of flavonoid biosynthesis, where the core pathway and genes are well characterized [1,3,4]. First, phenylalanine is converted to coumaroyl-CoA by phenylalanine ammonia lyase (PAL), cinnamate-4-hydroxylase (C4H), and 4-coumarate CoA ligase, which are conserved among higher plant. Second, dihydroflavonol is synthesized from one coumaroyl-CoA and three malonyl-CoA molecules and is catalyzed by many enzymes, such as chalcone synthase (CHS), chalcone isomerase (CHI), and flavanone 3-hydroxylase (F3H). A few branches have emerged in this key process. Third, dihydroflavonol is catalyzed to anthocyanidin by dihydroflavonol 4-reductase (DFR), and anthocyanidin synthase (ANS), flavonol by flavonol synthase (FLS), or proanthocyanidins by leucoanthocyanidin reductase. Then, a series of anthocyanin modifications are catalyzed by uridine diphosphate glucose-flavonoid glucosyltransferase (UFGT) and anthocyanin O-methyltransferase to form stable anthocyanins.

Moreover, the structural genes involved in anthocyanin synthesis are also regulated by many transcription factors (TFs), such as bZIP, MYB, bHLH, MADS-box, and WD40 [5,6]. The regulatory function of the MBW (MYB-bHLH-WD40) protein complex has been well established [7,8]. Environmental factors, such as UV radiation, temperature, light, and drought, can also affect anthocyanin formation [9,10,11,12].

The genus *Lilium* (family Liliaceae) is a bulbous monocot comprising more than 100 species [13]. As important cut and potted flowers, they exhibit many horticultural features, such as flower color, shape, and fragrance. The flower colors are principally red, pink, orange, yellow, white, and some intermediate colors. There are also some special colors, such as green lily (*L. fargesii*) and purple lily (*L. Souliei*). After years of breeding, many new commercial lilies have been cultivated, showing extensive variation in flower color [14]. The Royal Horticulture Society in the UK divided lily varieties into nine hybrids according to their characteristics. Among them, the typical characteristics of Asiatic hybrids include a wide variation in flower colors, from yellow to orange, pink, red, or white. The main flower colors of Oriental hybrids are pink, reddish violet, and white, while *L. longiflorum* hybrids are mainly white [15].

Lilies also have many variations in flower coloring patterns, such as bicolors and spots [15]. In bicolor lilies, different anthocyanin types and contents accumulate in different tepals parts; in *L. regale* and some trumpet lilies, anthocyanins accumulate in the throat. Some Asiatic hybrids have different colors in the upper and lower tepals. In Oriental hybrids and some of their parents, the anthocyanin coloring in the vascular bundle region of tepals is star-shaped, which is inconsistent with the other tepal parts.

In many wild lilies and lily hybrids, the inner surfaces of the tepals are blotched with red or dark spots. These spots are generally divided into three types. The first type is raised spots, located on the adaxial surface of tepals. The increase in epidermal and parenchymal cells can cause the surface of the tepals to uplift. Anthocyanins accumulate in both epidermal and parenchymal cells, and cell division may increase before pigment synthesis [15,16]. The second type is splatters: many small spots are splattered in the lower part of the tepals. The spot area is not uplifted and the surface is smooth. Anthocyanins only accumulate in epidermal cells, and there is no difference in morphology between pigmented cells and those without pigments [16,17]. The third type is brush marks: there include brush-like nut-brown stripes along the vascular bundles at the base of the tepals [15].

*L. leichtlinii* var. *maximowiczii* (section Sinomartagon), a wild species known for its tolerance of cold native to Japan, North Korea, the Far East of Russia, and north and northeastern China, is an important species used for breeding Asiatic hybrids [18]. Meanwhile, the bulbs of *L. leichtlinii* var. *maximowiczii* have been used as food and medicine in China, Japan, and Korea as typical edible lily species [19]. The flower is recurred and pendulous, and its petals are orange-red, with raised black spots. However, anthocyanins in the spots on the petals have not been sufficiently identified or quantified. In this study, transcriptomic and metabolomic technologies were used to identify differentially expressed genes (DEGs) and to reveal anthocyanin changes in petals that determine spot formation in *L. leichtlinii* var. *maximowiczii*. Through RNA-seq abundance and high-performance liquid chromatography (HPLC), we investigated the transcript profiles of the petal spots during multiple floral development stages to elucidate the relationship between structural genes and TFs for anthocyanin biosynthesis in *L. leichtlinii* var. *maximowiczii* petal spot. Our results provide new insights for the identification of functional genes and metabolites in spot formation, which lays a foundation for spot improvement using advanced breeding technology in lily varieties.

## 2. Results

### 2.1. Morphology Analysis of Petal Color Transitions and Spots Formation

The petal color of every flower transformed continuously from green to yellowish green, and then from light orange to orange–red during floral development in *L. leichtlinii* var. *maximowiczii*. According to the petal color transition and spot appearance processes, the floral developmental stages were described as follows: no obvious anthocyanin or carotenoid pigmentation was observed at stage 1 (D1), anthocyanin coloring at spots started to appear at stage 2 (D2), carotenoid coloring of the petal background was obvious at stage 3 (D3), full pigmentation of the petal background occurred at stage 4 (1 d before anthesis, D4), and flowers blossomed at stage 5 (D5) (Figure 1A). Morphological analysis showed no orange coloration in the petals before floral development stage D3. The raised spot formation at D2 (Figure 1C) was ahead of the petal color transitions at D3. Based on morphological anatomy analysis, we collected petals (Pe) and spots (Sp) at D1, D2, D3, and D4 for transcriptome sequencing and metabolome identification (Figure 1B).

### 2.2. Carotenoid Identification and Quantification in Petals

To obtain an accurate understanding of carotenoid accumulation in petals, carotenoid profiling was performed on *L. leichtlinii* var. *maximowiczii* petals using LC-ESI-MS/MS during petal color transitions. Fourteen carotenoids were detected in D1-Pe, D2-Pe, D3-Pe, and D4-Pe (Table 1). The major carotenoids in D3-Pe and D4-Pe were capsanthin, capsorubin, zeaxanthin, and violaxanthin. The levels of these four carotenoids showed an upregulation trend from D1 to D4. The capsanthin content in D1-Pe and D2-Pe was 1.22 µg/g and 2.27 µg/g, respectively. Then, the content drastically increased hundreds-fold to 406.27 µg/g in D3-Pe and 940.63 µg/g in D4-Pe. The capsorubin content significantly increased from 0.25 µg/g in D1-Pe and 1.65 µg/g in D2-Pe to 215.28 µg/g in D3-Pe and 493.43 µg/g in D4-Pe. The zeaxanthin and violaxanthin contents were 31.24 µg/g and 29.07 µg/g in D4-Pe, respectively, which were both much lower than the contents of capsanthin (940.63 µg/g) and capsorubin (493.43 µg/g) in D4-Pe.

Lutein had the highest carotenoid content in D1-Pe, which showed a decreasing trend. The lutein content slightly decreased from 158.63 µg/g in D1-Pe to 114.62 µg/g in D2-Pe and then significantly decreased to 32.10 µg/g in D3-Pe and 1.27 µg/g in D4-Pe.

### 2.3. Anthocyanin Identification and Quantification in Petal Spots

Flavonoids, particularly anthocyanins, are the predominant pigment molecules found in many flowers. To understand the changes in anthocyanin content in petal spots, anthocyanins were quantified using LC-ESI-MS/MS. In total, 40 anthocyanin-related compounds were detected, which may be responsible for the spot coloration (Figure 2). These 40 anthocyanins were classified into 10 categories as follows: cyanidins (13), delphinidins (5), flavonoids (5), malvidins (1), pelargonidins (4), peonidins (6), and petunidins (6) (Figure 3).

Most anthocyanins showed an upregulation trend along with malvidin-3-O-galactoside and pelargonidin-3-O-glucoside. In D1-Sp, malvidin-3-O-galactoside content was only 0.0034 µg/g, and it was undetectable in D4-Sp. Pelargonidin-3-O-glucoside was only detected in D2-Sp at a content of 2.84 µg/g (Appendix A).

### 2.4. Differentially Accumulated Anthocyanin Components in Petal Spots

To comprehensively screen the differentially accumulated metabolites (DAMs) between pairs of spot samples (D1-Sp vs. D2-Sp, D1-Sp vs. D3-Sp, and D1-Sp vs. D4-Sp) in *L. leichtlinii* var. *maximowiczii*, metabolites with variable importance in projection (VIP) value ≥1, fold change ≥2 or ≤0.5, and content ≥1 µg/g (at least one stage) were selected. The differences in metabolite composition and their expression levels, shown in a Venn diagram and heat map, indicate significant DAMs among the three comparisons, including 18 upregulated anthocyanins in D1-Sp vs. D2-Sp, D1-Sp vs. D3-Sp, and D1-Sp vs. D4-Sp (Figure 4A–C). The Venn diagram showed 17 common anthocyanins among the three comparisons, including cyanidin (7), delphinidin (3), pelargonidin (1), peonidin (2), petunidin (1), and flavonoids (3) (Figure 4D, Appendix A).

The 17 common anthocyanins included the top 10 most abundant anthocyanins, including cyanidin, pelargonidin, and peonidin, especially cyanidin-3-O-glucoside, pelargonidin-3-O-rutinoside, cyanidin-3-O-rutinoside, and peonidin-3-O-rutinoside, which play a significant role in spot formation in *L. leichtlinii* var. *maximowiczii* (Table 2). Anthocyanins identified in the three comparison groups showed similar accumulation, which corresponded to the spot color intensity of *L. leichtlinii* var. *maximowiczii*. We propose that the DAMs found in anthocyanin biosynthesis, especially the top ten highest anthocyanidin contents, may contribute to spot coloring in *L. leichtlinii* var. *maximowiczii* petals.

### 2.5. De Novo Assembly and Gene Function Annotation

RNA-seq analysis was performed to further study the molecular mechanism of spot formation. Twenty-four libraries were established using Pe and Sp at D1, D2, D3, and D4 stages (three biological replicates for each sample), and a total of 8.41 to 11.05 G clean base for each sample were obtained through sequencing. A total of 1,598,209,652 raw reads and 1,541,848,276 clean reads were obtained from twenty-four samples. The Q20 and Q30 values of each library were ≥97.93 and 94.17%, respectively. The GC content of each sample ranged from 49.37–50.79%, with an average of 50.22% (Appendix A).

A total of 277,166 unigenes were obtained, with an average length of 743 bp and an N50 of 1088 bp, using the Trinity (v2.11.0) method (Appendix A). The total number of unigenes was much higher than the other eukaryotic; we thought that there were quite a number of lncRNAs in the assembled unigenes. Unigenes without coding potential, as predicted by CNCI (version 2), CPC (version 0.9-r2), CPAT (1.2.4), and the intersection of the non-protein-coding potential results were chosen as novel long non-coding RNAs (lncRNAs). As a result, 151,202 (54.55%) unigenes were identified as lncRNAs, the remaining 125,964 (45.45%) unigenes were identified as coding unigenes, and our subsequently RNA-seq analysis mainly focused on the protein-coding unigenes.

In total, 103,052 unigenes (81.81% of the 125,964 coding unigenes) were annotated against at least one database using BLASTx (E-value < 1 × 10^−5^) or hmmscan (E-value <  1 × 10^−5^). Among these, 99,660 (79.12%), 98,640 (78.31%), 67,396 (53.50%), 57,257 (45.46%), and 54,580 (43.33%) were annotated in the Nr, Trembl, Swiss-Prot, KOG, and Pfam databases, respectively (Appendix A). Based on GO analysis, 77,890 (61.84%) unigenes were successfully annotated using GO assignments and categorized into three main categories: biological process, cellular component, and molecular function (Appendix A). Biological processes mainly focused on cellular and metabolic processes. Cellular components were mainly involved in the ‘cell’ and ‘cell part.’ The molecular functions were mainly classified into ‘binding’ and ‘catalytic activity.’ KEGG term analysis was used to identify the functional classifications of the unigenes. A total of 69,063 (54.83%) unigenes were enriched in 19 KEGG pathway groups, of which ‘carbohydrate metabolism’ represented the largest group, followed by ‘translation’ and ‘folding, sorting and degradation’ (Appendix A).

### 2.6. Identification of DEGs and Co-Expression Network Analysis

For each unigene, the fragment per kilobase of transcript per million mapped reads value was calculated to quantify the expression abundance and variation using RSEM software. Unigenes with a corrected *p* value < 0.05 and |log2 (fold change) | > 1 were considered DEGs. A total of 36,405 differentially expressed unigenes were identified from 16 comparisons, and the number of differentially expressed unigenes in each comparison is shown in Appendix A.

To obtain a comprehensive understanding of genes expressed in the petals and spots across the period of flowering and to identify the specific genes highly associated with spot formation, weighted gene co-expression network analysis was performed using the differentially expressed unigenes from 16 previously identified comparisons. Co-expression networks were constructed based on pairwise correlations of gene expression across all samples. Modules were defined as clusters of highly interconnected genes, and genes within the same cluster had high correlation coefficients. This analysis identified 22 distinct modules (labeled with different colors), as shown in the dendrogram in Figure 5A. Notably, three out of 22 co-expression modules, light yellow, dark green, and dark magenta, were composed of genes highly correlated with spot formation and anthocyanin accumulation at D2 and D3, with an upregulation trend. GO and KEGG enrichment analyses were subsequently performed for each module. The light yellow module consisted of 1618 genes. Their expression significantly increased in D2 and D3 and then decreased in D4. KEGG enrichment analysis showed significant enrichment of the flavonoid biosynthesis (ko00941) pathway, including 28 flavonoid-related unigenes, such as CHS, CHI, F3H, F3′H, and MT (Figure 5C; Appendix A). Among the 76 identified TFs in the light yellow module, 24 candidates were involved in anthocyanin accumulation, including MYB, bHLH, WD40, WRKY, bZIP, and ERF (Figure 5B; Appendix A).

A total of 318 genes were identified in the dark magenta module, displaying a modest increase from D1-Sp to D2-Sp, compared with the light yellow module, and the flavonoid biosynthesis (ko00941) pathway was the most enriched (Figure 5D). Thirteen flavonoid-related unigenes were included in this pathway: two C4H (cluster-12430.76377 and cluster-12430.145586), one DFR (cluster-12430.108267), four FLS (cluster-12430.139190, cluster-12430.145413, cluster-12430.107715, and cluster-12430.129622), two CHS (cluster-12430.148214 and cluster-12430.106213), one LDOX/ANS (cluster-12430.119154), two F3H (cluster-12430.120018 and cluster-12430.123007), and one ANR (cluster-12430.2911) (Appendix A). One of each MYB (cluster-12430.176730) and WRKY (cluster-12430.137802) gene was identified as a candidate TF associated with anthocyanin biosynthesis. A total of 606 genes were identified in the dark green module; only 3 genes were enriched in flavonoid biosynthesis (ko00941), i.e., MT (cluster-12430.119063), DFR (cluster-12430.106530), and LDOX/ANS (cluster-12430.119187).

### 2.7. Specifically Expressed Genes in Petal Spots

To identify genes that were specifically expressed in the spots, we performed analyses between spots (Sp) and petals (Pe) at D2 when purple spots emerged. Approximately 0.46% (1273 genes) of the total number of unigenes (125,964) were specifically expressed in the spots (D2-Sp) (Appendix A). We focused on subsequent analyses of unigenes putatively involved in flavonoid metabolism, particularly anthocyanin biosynthesis and transportation. In total, 27 genes and 11 TFs related to anthocyanin biosynthesis were identified, including CHS (10), DFR (3), CHI (2), PAL (2), MT (10), and MYB (6), bHLH (2), AP2-ERF (2), bZIP (1) (Appendix A).

### 2.8. Key DEGs Responsible for Anthocyanin Biosynthesis and Quantitative Reverse Transcription PCR (RT-qPCR) Validation

In total, enrichment and specific expression analysis identified 62 structural genes and 37 unigenes encoding seven TFs (MYB, WD40, bHLH, AP2/ERF, bZIP, WRKY, and zf-HD) responsible for anthocyanin biosynthesis (Appendix A). There were 62 DEGs encoding 9 enzymes in the flavonoid and anthocyanidin biosynthetic pathways (Figure 6A). The transcript abundance of seven key structural gene families, including CHS (20 DEGs), CHI (4 DEGs), F3H (3 DEGs), F3′H (2 DEGs), DFR (4 DEGs), LDOX/ANS (1 DEG), and MT (6 DEGs) were higher in Sp than in Pe at the D2 and D3 stages, consistent with the high abundance of anthocyanins and spot formation stage (Figure 6A).

Nine DEGs involved in anthocyanin biosynthesis were selected to confirm the RNA-seq results, and the expression levels were analyzed in Pe and Sp at four stages of floral development using RT-qPCR. Primer sequences for the genes are listed in Appendix A. Nine DEGs involved in anthocyanin biosynthesis were differentially expressed in D2-Sp and D3-Sp, including CHSA (cluster-12430.130661), CHSB (cluster-12430.119863), FLS (cluster-12430.116706), F3H (cluster-12430.123007), F3′H (cluster-12430.113781), LDOX/ANS (cluster-12430.119187), MYB12 (cluster-12430.132482), MYB (cluster-12430.86195), and MYB-related (cluster-12430.165019) (Figure 6B). The expression patterns of these DEGs corresponded well with the RNA-seq results.

## 3. Discussion

The main aim of this study was to understand the transcriptional control of spot formation and anthocyanin distribution in *L. leichtlinii* var. *maximowiczii* flowers. In Asiatic hybrids, carotenoids are the main color components of yellow and orange flowers, anthocyanin is predominant in pink and brown flowers, and red flowers contain both anthocyanins and carotenoids [20,21]. A previous study reported that lutein, violaxanthin, and β-carotene are major carotenoids in green tissues, which are conserved among plants [22]. In contrast, carotenoid compositions in flowers and fruits are different from those in green tissues, and large variations have been observed among species [15,23].

Previous studies have shown that the color components of orange petals, such as *L. amabile*, *L. davidii* var. *willmottiae*, and *L. leichtlinii* var. *maximowiczii* include capsanthin, capsorubin, zeaxanthin, violaxanthin, and antheraxanthin [24]. The orange color of tiger lily (*L. lancifolium* ‘Splendens’) flowers is primarily due to the accumulation of two k-xanthophylls: capsanthin and capsorubin [25]. Conversely, no anthocyanins have been detected in white Asiatic and Oriental hybrids and *L. longiflorum* [26,27,28]. Generally, the difference in the flower color of lily hybrids is caused by the difference in anthocyanin content; a large amount of cyanidin 3–O–β–rutinoside and a small amount of cyanidin 3–O–β–rutinoside–7–O–β–glucoside accumulate in some red varieties of Oriental and Asiatic hybrids [28]. The main component of spots on tepals is also cyanidin 3–O–β–rutinoside [29].

In our study, in the green petal at D1-Pe, the carotenoid with the highest content was lutein (158.63 μg/g), with small amounts of neoxanthin (13.47 μg/g), β-carotene (12.85 μg/g), and violaxanthin (11.66 μg/g) were also present. When the petal background turned light orange at D3-Pe and orange at D4-Pe, all carotenoids, except violaxanthin, showed a downward trend (Figure 1, Table 1). Capsanthin and capsorubin drastically increased at D3-Pe and D4-Pe during petal color transitions. A large amount of capsanthin and capsorubin, with small amounts of zeaxanthin and violaxanthin, may contribute to the orange color of *L. leichtlinii* var. *maximowiczii* petals.

There are six main anthocyanins in plants: cyanidin, delphinidin, pelargonin, peonidin, malvidin, and petunidin, which can change the color of flowers from pink to blue [1,30]. Anthocyanins are highly unstable; therefore, they are bound to glycosides to enhance their stability in organisms [31]. In our study, except for malvidin, the other five aglycones of anthocyanins were detected in the petal spots, namely cyanidin, pelargonidin, peonidin, petunidin, and delphinidin. Four major types of glycosides were identified, including two monoglycosides (glucoside and galactoside) and two polyglycosides (rutinoside and sophoroside). Purple spots emerged at D2-Sp, and cyanidin-3-O-glucoside drastically increased to 195 μg/g from 0 μg/g at D1-Sp. When the spots turned dark, the cyanidin-3-O-glucoside content increased gradually. Besides cyanidin-3-O-glucoside, as previously reported [29], pelargonidin-3-O-rutinoside, cyanidin-3-O-rutinoside, and peonidin-3-O-rutinoside may also be major contributors to the spot color of *L. leichtlinii* var. *maximowiczii* petals.

Anthocyanins are synthesized through a series of enzymatic reactions along the phenylalanine pathway and are further modified by glycosylation, methylation, and acylation until they are finally stored in vacuoles [32]. Anthocyanin biosynthesis gene expression levels determine the anthocyanin content in plants [33]. In a recent study, eight structural genes (CHSa, CHSb, CHIa, CHIb, F3H, F3′H, DFR, and ANS) and three TF genes (MYB12, MYB15, and bHLH2) were isolated, and anthocyanin accumulation was examined during flower development in four cultivars of two *Lilium* species. High transcript accumulations were observed in the red tepals of ‘Gran Tourismo,’ followed by the pink tepals of ‘Perth,’ and both tepals contained cyanidin. The white tepals of ‘Rialto’ and ‘Lincoln’ did not contain anthocyanin and showed the lowest transcript accumulations [34]. The expression of these structural genes is strongly correlated with MYB12 and MYB15 expression [34]. Three CHS genes and one DFR gene were cloned in the Asiatic hybrid ‘Montreux.’ LhCHSA and LhCHSB were expressed in the tepals, filaments, and pistils. LhCHSC is specifically expressed in anthers [35,36]. DFR is only expressed in the organs where anthocyanin accumulates, whereas CHS is also expressed in some sites where anthocyanin does not accumulate, indicating that DFR is more closely related to anthocyanin coloring than CHS in Asiatic lilies. Coloring analysis of the filial generation showed that CHS and DFR independently control tepal and spot coloring [35].

In our study, seven key structural genes were activated during anthocyanidin biosynthesis in *L. leichtlinii* var. *maximowiczii*. Among them, the expression of DEGs encoding CHS, CHI, F3H, F3′H, DFR, and ANS was higher in D2-Sp and D3-Sp than in D2-Pe and D3-Pe, which may result in higher anthocyanidin levels during spot formation (Figure 6). Twenty CHSs, including CHSA, CHSB, and CHSC, were significantly upregulated in spots (D2-Sp, D3-Sp, and D4-Sp), but could not be detected in petals (D1, D2, D3, D4-Pe), as the RNA-seq results showed (Figure 6B; Appendix A). CHS is a key enzyme in anthocyanin biosynthesis, and the loss of its activity typically results in albino flowers [37,38]. Similarly, in the tree peony cultivar ‘Qing Hai Hu Yin Bo’, PsCHS was strongly and specifically expressed in petal blotches that appeared at the base of the white petals [39], which was most likely the direct cause of anthocyanin blotch formation. Three CHI, three DFR, three ANS, one F3H, and two F3′H genes were identified in RNA-seq, showing the same expression profiles as CHS.

Although our study and other studies have clarified some key genes associated with flower coloration in *Lilium*, the relationship between gene expression and flower color development needs to be more thoroughly investigated. In many plants, additional genes have been identified as important factors that affect anthocyanin synthesis in flowers and fruits. In *Petunia hybrida* petals, CHS silencing inhibited the accumulation of anthocyanins and the mRNA levels of the corresponding endogenous targets, such as CHI and DFR were unaffected [40]. Sequence-specific degradation of CHS mRNA in petal sectors along the central veins is the known cause of the *P. hybrida* ‘Red Star’ pigmentation pattern [41]. In Lycium, CHS, CHI, F3′5′H, DFR, ANS, and UFGT expression levels were highly consistent with anthocyanin accumulation in black and red fruit [42,43]. In purple-red *Padus virginiana L.* leaves, the flavonoid biosynthesis genes (PAL, CHS, and CHI) and their transcriptional regulators (MYB, HD-Zip, and bHLH) exhibited increased expression during purple-red periods [44]. These findings indicate the importance of CHS, CHI, ANS, F3H, F3′H, DFR, and UFGT in anthocyanin biosynthesis.

In addition to key structural genes, TFs, such as MYB, bHLH, WD, and WRKY, also play important roles in anthocyanin biosynthesis [7,31]. In many *Lilium* species, MYB12 usually interacts with bHLH2 to form a MYB12/bHLH2 complex that upregulates structural gene transcription [28,45]. By inhibiting the transcription of MYB12, for example, in high heat (35 °C), the transcription accumulation of synthetic genes decreases, and the anthocyanin accumulation is thus reduced [9]. Two TFs are involved in regulating anthocyanin synthesis in the Asiatic hybrid ‘Montreux’ [46]. In the white tepals of ‘Rialto’ and *L. speciosum*, the W-to-L substitution in the R2 repeat of MYB12 was responsible for the absent transcriptional activation of anthocyanin structural genes, resulting in a lack of anthocyanin accumulation [27,47]. In *Clarkia gracilis*, a species with petal spots/blotches similar to tree peony, allelic variation in the cis-regulatory region of the R2R3-MYB gene, CgMYB1, restricts its expression and subsequent anthocyanin accumulation to either a basal or central petal spot/blotch in different subspecies [48]. Further, an R2R3-MYB transcription factor (CgsMYB12) was identified to be responsible for anthocyanin pigmentation of the basal region (‘cup’) in the petal of *C. gracilis* ssp. *sonomensis*. Additionally, two R2R3-MYB genes, CgsMYB6 and CgsMYB11, were involved in petal background pigmentation [49]. In this study, two MYB (cluster-12430.165019 and cluster-12430.132482 MYB12) and one bHLH (cluster-12430.66365) genes were specifically expressed in spots between D2-Sp and D2-Pe (Appendix A), and they were also enriched in the light yellow module (Appendix A). They may positively regulate the expression of most structural genes involved in flavonoid biosynthesis with the MBW complex during spot formation. In the Oriental hybrid lily ‘Sorbonne’, MYB12 regulates both whole tepal and raised spot pigmentation [50]. However, MYB12 (cluster-12430.132482) was specifically expressed in spots between D2-Sp and D2-Pe and D3-Sp and D3-Pe. At the D3 stage, the petals appeared light orange in color, indicating that more MYB was involved in petal coloration in *L. leichtlinii* var. *maximowiczii* flowers. Some TFs, such as AP2/ERF, bZip, and WRKY, display differential expression during spot formation, suggesting that they may synergistically regulate the expression of genes involved in flavonoid biosynthesis, as mentioned in a recent report, where the MYB and HD-Zip TFs exhibited similar expression, possibly acting as key regulators of flavonoid biosynthesis.

## 4. Materials and Methods

### 4.1. Plant Materials and Sampling

Wild lily *L. leichtlinii* var. *maximowiczii* species were used in this study. Bulbs were planted in pots on 3 March, and grown in a greenhouse (unheated) at the Shanghai Academy of Agricultural Sciences, Shanghai, China under natural light with daytime and nighttime temperatures of 17–26 °C. On 22 April, the flower buds were obvious. The floral developmental stages are described as follows, D1 (green petals without spots, bud length <3 cm), D2 (yellowish green petals with purple spots, bud length <3–4.5 cm), D3 (light-orange petals with dark-purple spots), D4 (orange petals with dark-purple spots, one day before anthesis), and D5 (the day of anthesis) (Figure 1A). Petals and spots were collected from flowers at D1, D2, D3, and D4 (Figure 1B). All materials were sampled, immediately frozen in liquid nitrogen, and stored at −80 °C.

### 4.2. Sample Preparation and Extraction

For Flavonoids: the spots samples were freeze-dried, ground into powder (30 Hz, 1.5 min), and stored at −80 °C until needed. Fifty milligrams of powder were weighed and extracted with 0.5 mL methanol/water/hydrochloric acid (500:500:1, *v*/*v*/*v*). Then, the extract was vortexed for 5 min, ultrasonicated for 5 min, and centrifuged at 12,000× *g* at 4 °C for 3 min. The residue was re-extracted by repeating the aforementioned steps under the same conditions. The supernatants were collected and filtered through a membrane filter (0.22 μm, Anpel) before LC-MS/MS analysis.

For carotenoids: the petals samples were freeze-dried, ground into powder (30 Hz, 1.5 min), and stored at −80 °C until needed. Fifty milligrams of powder was weighed and extracted with a 0.5 mL mixed solution of n-hexane: acetone: ethanol (1:1:1, *v*/*v*/*v*), 10 μL of internal standard mixed solution (20 μg/mL) was added into the extract as internal standards (IS). The extract was vortexed for 20 min at room temperature. The supernatants were collected after centrifuging at 12,000 r/min for 5 min at 4 °C. The residue was re-extracted by repeating the above steps again. To the supernatants, 0.5 mL saturated sodium chloride solution was added and vortexed, and the upper layer was collected, this step was repeated two times more. Then, the supernatant was evaporated to dryness and dissolved in 0.5 mL MTBE, then 0.5 mL 10% KOH-MeOH was added, the mixture was vortexed, and the reaction was allowed to take place at room temperature overnight. After the reaction, 1 mL saturated sodium chloride solution and 0.5 mL MTBE were added and vortexed, and the upper layer was collected, this step was repeated two times and the supernatant was evaporated to dryness and reconstituted in 100 μL mixed solution of MeOH/MTBE (1:1, *v*/*v*). The solution was filtered through a 0.22 μm membrane filter (0.22 μm, Anpel) for further LC-MS/MS analysis.

### 4.3. Metabolite Profiling

Flavonoids in the spots (D1-Sp, D2-Sp, D3-Sp, and D4-Sp) and carotenoids in the petals (D1-Pe, D2–Pe, D3–Pe, and D4–Pe) of flowers at the four stages were identified and quantified using an LC-ESI-MS/MS system (HPLC, UFLC SHIMADZU CBM20A system, www.shimadzu.com.cn/, accessed on 23 August 2021; MS, Applied Biosystems 6500 QTrap, www.appliedbiosystems.com.cn/, accessed on 23 August 2021). Three biological samples were evaluated for each part of the spots or petals (i.e., samples of the same segment from at least five individuals were mixed for one replicate) for a total of twenty-four samples.

Metabolites were identified and quantified by Wuhan MetWare Biotechnology Co., Ltd. (Wuhan, China) based on the self-built MWDB database and a public database. Carotenoids and flavonoids were analyzed using scheduled multiple reaction monitoring (MRM). Data acquisitions were performed using Analyst 1.6.3 software (Sciex). All metabolites were quantified using Multiquant 3.0.3 software (Sciex). The identified metabolites with significant differences in content were set with 0.5 ≥ fold change or ≥ 2, *p* value < 0.05, and VIP ≥ 1. To study the accumulation of specific metabolites, principal component analysis and orthogonal partial least squares-discriminant analysis were performed using R (www.r-project.org/, accessed on 23 August 2021).

### 4.4. RNA Extraction, Quantification, and Sequencing

Petals and spots from flowers at D1, D2, D3, and D4 were collected, and three independent biological replicates were used. The amount of 0.1 g of flower tissue was used for RNA extraction. Purity, quantification, and integrity of total RNA were estimated using the NanoPhotometer^®^ spectrophotometer (IMPLEN, Westlake Village, CA, USA), Qubit^®^ RNA Assay Kit in Qubit^®^ 2.0 Fluorometer (Life Technologies, Carlsbad, CA, USA), and the RNA Nano 6000 Assay Kit of the Bioanalyzer 2100 system (Agilent Technologies, Santa Clara, CA, USA). mRNA was purified from a total of 1 µg of RNA per sample using poly-T oligo-attached magnetic beads. A NEB fragmentation buffer was added to break the RNA into short segments, and the first-strand cDNA was synthesized by reverse transcriptase. The second-strand cDNA synthesis was subsequently performed using DNA Polymerase I and RNase H. In order to select cDNA fragments of preferentially 250~300 bp in length, the library fragments were purified with the AMPure XP system (Beckman Coulter, Beverly, USA). Finally, 24 libraries representing the collected petal and spot samples were prepared following standard procedures of the Illumina HiSeq^TM^ 4000 system (San Diego, CA, USA) according to the manufacturer’s instructions. Next, cDNA libraries were sequenced on an Illumina sequencing platform and 150 bp paired-end reads were generated by Metware Biotechnology Co., Ltd. (Wuhan, China).

### 4.5. Transcriptome Data Analysis

Raw data were cleaned by removing adaptors and low-quality reads using Fastp (v0.19.3) with default parameters. Transcripts were assembled using Trinity (v2.11.0), and then Corset was used to regroup relevant transcripts into ‘gene’ clusters (https://github.com/trinityrnaseq/trinityrnaseq, accessed on 30 August 2021). In total, 277,166 unigenes were assembled.

The resulting unigenes were evaluated by identifying the complete BUSCO hits [51]. Following gene annotation, the potential transcript sequences encoded by each gene were subjected to DIAMOND or HMMER analysis against the NCBI non-redundant protein, eggNOG, Swissprot, Trembl, and Pfam databases to identify homologous proteins in other species. Possible GO terms in the protein sequences were identified using SwissProt and Trembl database ID mapping. The genes were annotated with KEGG terms using KOBAS (version 3.0), and TF analysis was performed using iTAK. TransDecoder software (https://github.com/TransDecoder/TransDecoder/wiki, accessed on 30 August 2021) was used to predict the coding regions within the transcript sequences. Unigenes without functional annotation were used to perform lncRNA prediction using CPC (v0.9-r2), CPAT (v1.2.4), and CNCI (version Feb 2 28, 2014). Unigenes predicted as lncRNAs using all three software programs were identified as lncRNAs, and a total of 151,202 lncRNAs were identified.

For each sample, paired-end reads were mapped back to the unigenes using bowtie (v2.3.4.1). The rsem–calculate–expression function within the software package RSEM (v1.3.1) was used to estimate the expression levels for each gene using the expectation-maximization algorithm. The expression profiles generated for each sample were combined and used to detect DEGs using DESeq2 (v1.22.2), and unigenes with fold change values > 2 and an adjusted *p* value < 0.05, according to the False Discovery rate method of Benjamini and Hochberg, were selected as DEG candidates. Subsequently, the WGCNA package (v1.71) was used for co-expression network analysis. All software parameters used in this study are listed in Appendix A.

### 4.6. Petal Surface Analysis Using Scanning Electron Microscopy

The morphological characteristics of *L. leichtlinii* var. *maximowiczii* petals were examined using scanning electron microscopy. Samples were handled according to Salehi et al. [52] with minor modifications. The fresh petals and spots from the four stages were dissected at approximately 5 mm, fixed in 2.5% glutaraldehyde solution at room temperature for 2 h, and then washed five times for 10 min each in phosphate buffer (0.1 M), pH 7.4. Afterward, the samples were dehydrated by washing in a series of ethanol solutions [2 × 50% (30 min), 75% (30 min), 90% (30 min), 95% (30 min), and 2 × 100% ethanol (30 min)]. Then, the samples were put in isoamyl acetate for 20 min and critical point dried in an HCP-2 Hitachi (Tokyo, Japan) equipment, wherein they were placed first in a 50–80% liquid carbon dioxide (L-CO_2_) at 10 °C for 20 min, and then at 40 °C for 5 min. For coating with 10 nm of gold, the dried samples were placed in metal stubs and placed in an ion sputter (MC1000, Hitachi, Tokyo, Japan). A scanning electron microscope (SU8100, Hitachi, Tokyo, Japan) was used for observations.

### 4.7. RT-qPCR Analysis

To validate the transcriptome data, the relative expression of nine DEGs identified in the transcriptome analysis was evaluated via RT-qPCR using three biological and three technical replicates. Total RNA was extracted using TRIzol reagent according to the manufacturer’s instructions (Aidlab Biotechnology Co., Ltd., Beijing, China). The first-strand cDNA was synthesized using 2 μg of total RNA as a template with a first-strand cDNA Synthesis Kit (Toyobo, Japan) in a 20 μL reaction volume. Gene-specific primers for RT-qPCR were designed according to the selected sequences derived from RNA-seq (Appendix A). The 18 S rRNA gene was used as an internal control. RT-qPCR was performed using the TaKaRa SYBR Green Mix kit (TaKaRa, Beijing, China) and the ABI 7500 fast real-time detection system. The RT-qPCR program was initiated with a denaturation step (95 °C for 30 s), followed by 40 cycles of PCR (95 °C for 15 s, 58 °C for 30 s), melting (95 °C for 15 s, 60 °C for 1 min, 95 °C, 15 s). Relative expression analyses of quantitative data were performed using the 2^−∆∆Ct^ method [53].

## 5. Conclusions

In this study, metabolomic and transcriptome analyses were used to identify the key anthocyanins and genes responsible for *L. leichtlinii* var. *maximowiczii* spot coloration in the petals. Overall, 40 anthocyanin metabolites were detected in *L. leichtlinii* var. *maximowiczii* petal spots. Cyanidin-3-O-glucoside, pelargonidin-3-O-rutinoside, and cyanidin-3-O-rutinoside are the main anthocyanin components in *L. leichtlinii* var. *maximowiczii* spots. Moreover, nine enzymes (especially CHS and DFR) in the anthocyanin biosynthesis pathway and seven differentially expressed TFs (especially MYB12 and bHLH) were identified as candidate regulators contributing to the color diversity. The results of this study provide valuable information and new insights for the further evaluation of *L. leichtlinii* var. *maximowiczii* genetic diversity.

## Figures and Tables

**Figure 1 ijms-24-01844-f001:**
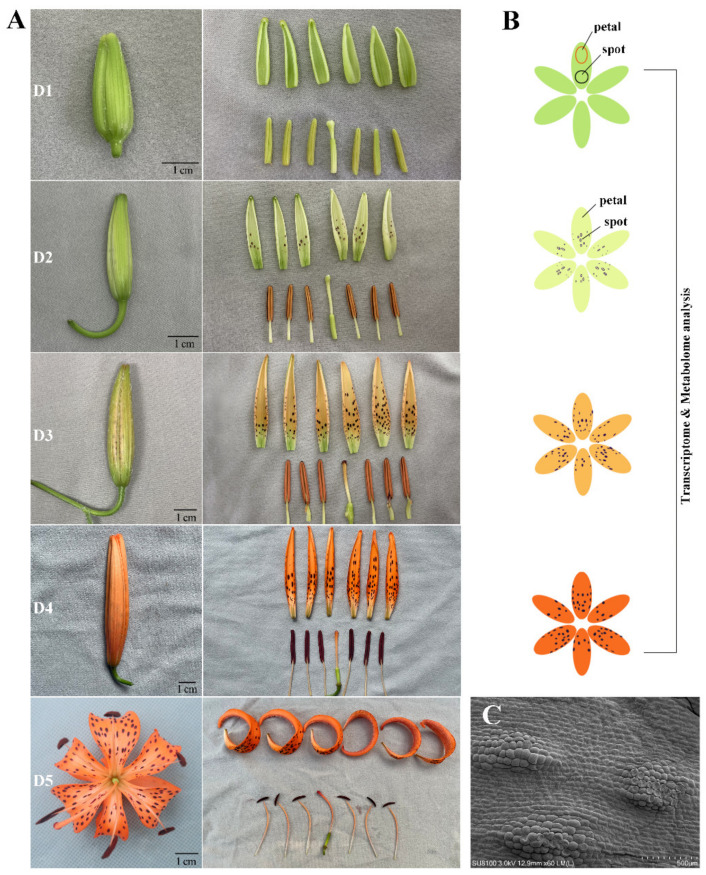
Morphological observation of *L. leichtlinii* var. *maximowiczii* flowers. (**A**) Morphological anatomy of *L. leichtlinii* var. *maximowiczii* flowers during floral development. Development stages D1 (green petals without spots), D2 (yellowish green petals with purple spots), D3 (light-orange petals with dark-purple spots), D4 (orange petals with dark-purple spots, one day before anthesis), and D5 (the day of anthesis). Bar = 1 cm. (**B**) Sampling strategy: petals and spots were collected at D1, D2, D3, and D4 for transcriptome sequencing and metabolome identification. (**C**) Scanning electron microscopy appearance of raised spots on D2 petals. Bar = 500 µm.

**Figure 2 ijms-24-01844-f002:**
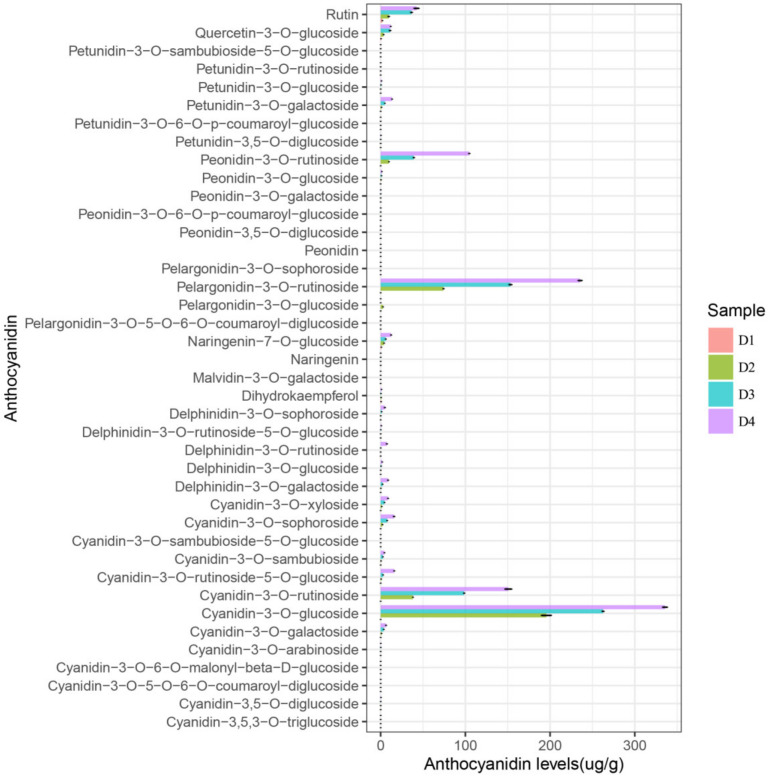
The anthocyanin content detected in different samples in this study. The *x*-axis represents the anthocyanin levels (µg/g). The *y*-axis represents the anthocyanin composition obtained using high-performance liquid chromatography (HPLC). Error bars show the SD of the mean. D1-Sp, lower part of green petals at stage 1; D2- Sp, D3- Sp, D4- Sp, spots on petals at different stages.

**Figure 3 ijms-24-01844-f003:**
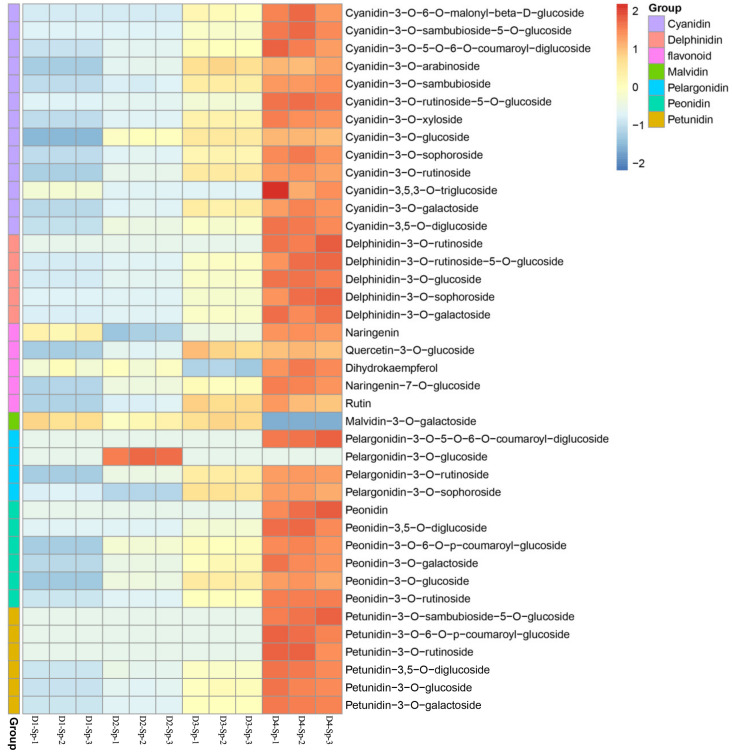
Heatmap of metabolites related to cyanidin, delphinidin, flavonoid, malvidin, pelargonidin, peonidin, and petunidin in D1-Sp, D2-Sp, D3-Sp, and D4-Sp. The marker on the right side of the heatmap represents the name of each anthocyanin composition obtained via HPLC.

**Figure 4 ijms-24-01844-f004:**
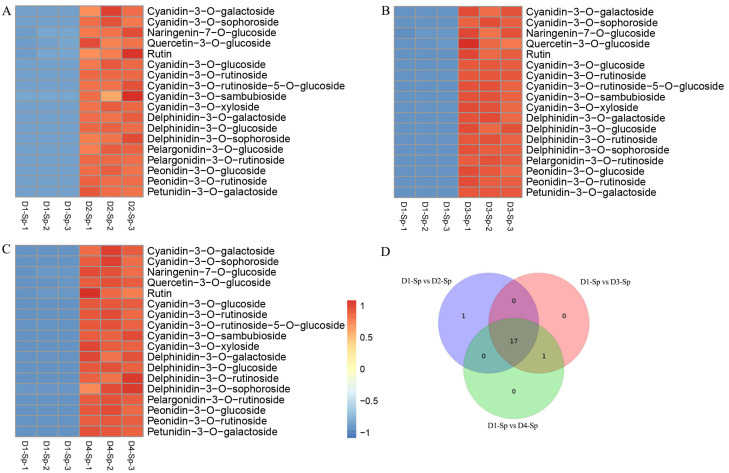
Differentially accumulated metabolite (DAM) analysis of the metabolome. Heatmaps of DAMs in D1-Sp vs. D2-Sp (**A**), D1-Sp vs. D3-Sp (**B**), and D1-Sp vs. D4-Sp (**C**). (**D**) Venn analysis of D1-Sp vs. D2-Sp, D1-Sp vs. D3-Sp, and D1-Sp vs. D4-Sp. The color scale from Min (blue) to Max (red) indicates the metabolite contents from low to high. Identification of DAMs between three comparison groups was performed using variable importance in projection values ≥1 and fold change ≥2 or ≤0.5; in addition, the content must be ≥ 1 μg/g DW in spots in at least one stage.

**Figure 5 ijms-24-01844-f005:**
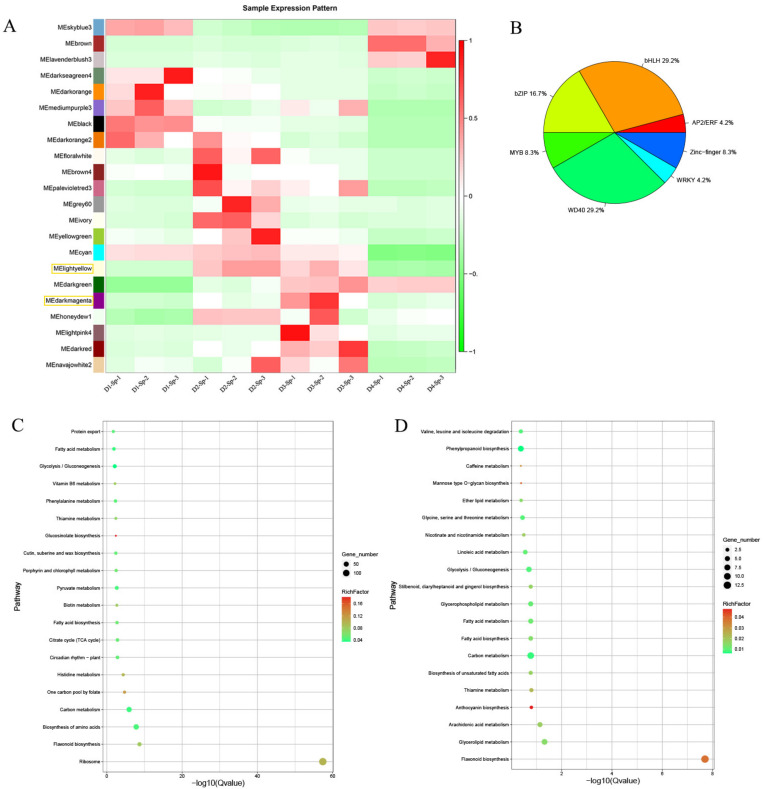
Gene regulation during spot formation from D1 to D4. (**A**) K-means cluster analysis of co-expressed genes and their expression patterns. (**B**) Differentially expressed genes (DEGs) involved in transcription factor enrichment in the light yellow module. (**C**) KEGG enrichment bar plot of DEGs in the light yellow module. The light yellow cluster represents the expression pattern of 1618 co-expressed genes identified via K-means cluster analysis. (**D**) KEGG enrichment bar plot of DEGs in the dark magenta module including 318 identified co-expressed genes.

**Figure 6 ijms-24-01844-f006:**
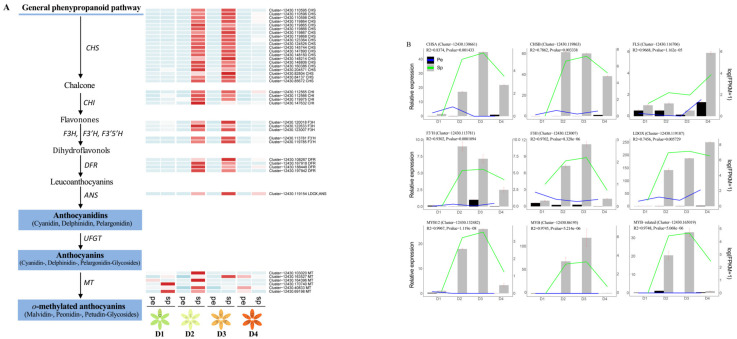
Analysis of DEGs in the anthocyanin biosynthesis pathway in spots on *L. leichtlinii* var. *maximowiczii* petals. (**A**) Reconstruction of the anthocyanin biosynthetic pathway with the structural DEGs. The DEGs were identified using an adjusted *p* value < 0.05 and |log2 fold change| ≥ 1. (**B**) Validation of the expression of anthocyanin-related genes in Pe and Sp at the four stages using RT-qPCR. Error bars indicate the SD of three independent biological repeats.

**Table 1 ijms-24-01844-t001:** Contents (μg/g) of carotenoids in D1-Pe, D2-Pe, D3-Pe, and D4-Pe.

Class	Compounds	Molecular Weight (Da)	Q1 (Da)	Q3 (Da)	Rt (Min)	D1-Pe	D2-Pe	D3-Pe	D4-Pe	Trend
Carotenes	β-carotene	536.44	537.60	177.10	6.23	12.85 ± 0.31 ^a^	11.82 ± 0.07 ^b^	4.93 ± 0.10 ^c^	1.22 ± 0.06 ^d^	down
	(E/Z)-phytoene	544.50	545.30	81.00	4.83	0.00 ^c^	1.92 ± 0.15 ^b^	0.00 ^c^	4.81 ± 0.80 ^a^	-
Xanthophylls	capsanthin	584.87	585.50	109.10	4.47	1.22 ± 0.07 ^c^	2.27 ± 0.15 ^c^	406.27 ± 33.40 ^b^	940.63 ± 98.30 ^a^	up
	capsorubin	600.42	601.40	109.00	4.36	0.25 ± 0.01 ^c^	1.65 ± 0.04 ^c^	215.28 ± 11.42 ^b^	493.43 ± 51.11 ^a^	up
	zeaxanthin	568.43	569.40	477.50	4.61	0.85 ± 0.02 ^c^	2.12 ± 0.16 ^c^	10.73 ± 0.46 ^b^	31.24 ± 1.81 ^a^	up
	violaxanthin	600.42	601.40	221.00	1.59	11.66 ± 0.32 ^c^	6.89 ± 0.36 ^d^	13.11 ± 0.58 ^b^	29.07 ± 0.72 ^a^	-
	β-cryptoxanthin	552.43	553.50	177.40	5.49	1.66 ± 0.34 ^c^	1.73 ± 0.34 ^c^	2.29 ± 0.22 ^b^	3.74 ± 0.10 ^a^	up
	lutein	568.43	551.50	175.40	4.00	158.63 ± 9.98 ^a^	114.62 ± 1.51 ^b^	32.10 ± 1.36 ^c^	1.27 ± 0.09 ^d^	down
	neoxanthin	600.42	601.40	565.50	1.96	13.47 ± 0.10 ^a^	11.45 ± 0.84 ^b^	8.25 ± 0.76 ^c^	5.55 ± 0.43 ^d^	down
	antheraxanthin	584.42	585.50	175.40	2.87	2.05 ± 0.07 ^b^	2.20 ± 0.13 ^b^	10.99 ± 1.07 ^a^	0.00 ^c^	-
	α-cryptoxanthin	552.43	553.50	123.10	5.04	0.16 ± 0.04 ^ab^	0.00 ^c^	0.11 ± 0.01 ^b^	0.19 ± 0.03 ^a^	-
	8′-apo-beta-carotenal	416.64	417.30	325.30	4.46	0.00 ^b^	0.00 ^b^	0.00 ^b^	0.04 ± 0.00 ^a^	-
	echinenone	550.90	551.60	203.10	5.51	0.01 ± 0.00 ^a^	0.01 ± 0.00 ^a^	0.00 ± 0.00 ^b^	0.01 ± 0.00 ^a^	-
	β-citraurin	432.60	433.30	341.10	2.79	0.00 ^b^	0.00 ^b^	0.05 ± 0.00 ^b^	0.61 ± 0.07 ^a^	-

Data are expressed as the means ± standard deviation (SD) of three biological replicates. Different letters indicate significant differences at *p* < 0.05 (Duncan’s multiple range test).

**Table 2 ijms-24-01844-t002:** The top ten highest anthocyanidin contents (μg/g) in D4-Sp.

Compounds	Molecular Weight (Da)	Q1(Da)	Q3 (Da)	Rt (Min)	D1-Sp	D2-Sp	D3-Sp	D4-Sp	Trend
Cyanidin-3-O-glucoside	449.11	449.10	287.10	5.54	0.00 ^d^	195.67 ± 7.04 ^c^	262.56 ± 1.27 ^b^	335.65 ± 3.37 ^a^	up
Pelargonidin-3-O-rutinoside	579.17	579.06	271.10	7.25	0.00 ^d^	74.23 ± 0.55 ^c^	153.24 ± 1.96 ^b^	235.45 ± 2.71 ^a^	up
Cyanidin-3-O-rutinoside	595.17	595.17	287.10	6.16	0.00 ^d^	38.23 ± 0.13 ^c^	98.59 ± 0.61 ^b^	150.75 ± 4.84 ^a^	up
Peonidin-3-O-rutinoside	609.19	609.50	301.10	7.87	0.00 ^d^	9.84 ± 0.14 ^c^	39.20 ± 0.69 ^b^	104.65 ± 0.58 ^a^	up
Rutin	610.15	611.20	303.10	8.88	2.07 ± 0.09 ^d^	9.57 ± 0.81 ^c^	36.45 ± 1.08 ^b^	42.36 ± 3.48 ^a^	up
Cyanidin-3-O-rutinoside-5-O-glucoside	757.22	757.22	287.10	4.06	0.00 ^d^	0.67 ± 0.03 ^c^	2.95 ± 0.07 ^b^	16.06 ± 0.34 ^a^	up
Cyanidin-3-O-sophoroside	611.16	611.20	287.15	4.90	0.00 ^d^	2.17 ± 0.10 ^c^	7.76 ± 0.19 ^b^	15.83 ± 0.75 ^a^	up
Petunidin-3-O-galactoside	479.12	479.10	317.10	5.88	0.00 ^d^	1.06 ± 0.04 ^c^	5.03 ± 0.02 ^b^	13.52 ± 0.22 ^a^	up
Naringenin-7-O-glucoside	434.12	435.10	273.10	9.39	0.79 ± 0.06 ^d^	4.04 ± 0.18 ^c^	6.11 ± 0.28 ^b^	12.38 ± 0.43 ^a^	up
Quercetin-3-O-glucoside	464.10	465.10	303.10	8.79	0.46 ± 0.03 ^c^	3.50 ± 0.20 ^b^	11.07 ± 0.87 ^a^	11.94 ± 0.19 ^a^	up

Data are expressed as the means ± SD of three biological replicates. Different letters indicate significant differences at *p* < 0.05 (Duncan’s multiple range test).

## Data Availability

Transcriptome sequencing data are available in the National Genomics Data Center under accession number PRJCA013759.

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
