# Peer review of "Molecular and Metabolic Insights into Anthocyanin Biosynthesis for Spot Formation on Lilium leichtlinii var. maximowiczii Flower Petals"

_ijms, 2023, doi:10.3390/ijms24031844_

Round 1

Reviewer 1 Report

The authors have written the manuscript very well. Furthermore, the manuscript contains enough data to publish in the journal of IJMS. However, the manuscript has some minor errors (abbreviations and English). Therefore, authors should recheck the entire manuscript and rectify the errors. Furthermore, the authors may add some details in the materials and methods section. For example, how many days old tissue was collected for RNA isolation? Authors can mention the concentration of primer, SYBR mix and dilutions of cDNA used in the materials and methods section. How much Total RNA was used for cDNA synthesis? Could you mention the specific temperature and product size of qRT-PCR primers?

Author Response

(1) Some minor errors (abbreviations and English) were corrected in the revised manuscript. (2) The sample collection and RT-qPCR details were added in part 4.1 and 4.7 of the revised manuscript, respectively. (3) The length of products has been added in the suppl table S13, and the RT-qPCR primers’ annealing temperature was between 55-60 oC. And the 58 oC was used when running the programme (part 4.7).

Reviewer 2 Report

Authors describe diversity of anthocyanins and try to deduce petal spot formation and coloration in Lily by comparing metabolite and transcript accumulation data

Why results from D5-Sp are not presented as one can understand from figure 1, it seems to be a mature flower.

More information regarding collection of flower tissue, growth conditions of the plant may be presented in material n methods. This may help the reader understand. Influence of light dependent biosynthesis of pigments are not discussed. Did the authors observe any changes in spot formation under different growth conditions? How important is light for pigment accumulation ln lily?

Quantification of anthocyanin compounds and carotenoids has been presented. Was it quantified against appropriate standards. If yes, why is it not mentioned in methodology section.

Was the profiling of metabolites done using any specialized microdissection method or from whole tissue. Authors write metabolite profiling from spots on petal tissue showed presence of carotenoids etc. How was this carried out?

Authors have carried out identification of lncRNA's however, not much  discussed in results and discussion. Authors must elaborate and change the sections/highlight results regarding this. 

Author Response

(1) D4 is the one day before flowering, D5 is the day of flowering, color showed no obvious different, so we collected petals and spots samples at D4 stage. The flower of D5 placed in Figure1 was rather to show the mature flower characteristics of L. leichtlinii var. maximowiczii.

(2) The plant time, growth conditions and sample collection details were added in part 4.1 of the revised manuscript.

(3) In the daily planting, we did find that weak light or continuous rainy days would affect the coloring of petals and spots, which would make both colors lighter. In this experiment, we started to plant bulbs in March and began to collect samples in Mid-April. Both temperature and light were suitable for the growth of lily. For very few single flowers with lighter colors, we did not sample them.

(4) The more details related to metabolome was added in the part 4.2 and 4.3 of the revised manuscript.

(5) In this experiment, flavonoids in the spots (D1-Sp, D2-Sp, D3-Sp, and D4-Sp) and carotenoids in the petals (D1-Pe, D2–Pe, D3–Pe, and D4–Pe) of flowers at the four stages were identified and quantified. The dark spots emerged at D2 stage, it is easy to sample petals and spots individually at D2, D3 and D4 stages (Figure 1). At D1 stage, there are macroscopical tiny white spot at the lower part of petals which would turn to the dark spots later at D2. So, we sample upper and lower part of petals as the D1-pe and D1-Sp, respectively.

 (6) Thanks for pointing out this problem.  We agree that the analysis of lncRNAs were not explained clearly. As Show in Table S5, a total of 277,166 unigenes were assembled, of which only 103,052(37.18%) unigenes can be annotated in public databases. The total number of unigenes is quite higher than the other eukaryotic, and the percentage of annotated unigenes in public databases is quite low. As we know, lncRNAs with poly-A tails may be captured by poly-T oligo when preparing the library, we thought the lncRNAs were the cause of the problems discovered previously. So, the identification of lncRNAs were done and the 277,166 unigenes were classified into 151,202 lncRNAs (54.55%) and 125,964 protein coding mRNAs (45.45%), thus the number of protein coding unigenes and the percentage of annotated unigenes in public databases reached the normal level. The subsequently analysis were focus on protein coding unigenes, no much discussion on lncRNAs were processed. The purpose of lncRNAs identification were added to the text part 2.5 of the revised manuscript.

Reviewer 3 Report

Attachment

Author Response

  • The contend of information related to the economic value of Lilium leichtlinii and how breeding was added in the introduction of the the revised manuscript (Line 81-85).
  • The plant time, sample collection details and D5 description were added in part 4.1 of the revised manuscript.
  • Some other errors (abbreviations and English) like Line 29, 169, etc, were all corrected in the revised manuscript.
  • “The sequencing method with fragment size selection” was added in part 4.4 (Line 473-483) of the revised manuscript.

Round 2

Reviewer 2 Report

Accepted